# Calcium phosphate formation and deposition in ischemic neurons

**Shu Q. Liu**[1]*, **John B. Troy**[1], **Jeremy Goldman**[2], **Roger J. Guillory**[1]¤

**1** Biomedical Engineering Department, Northwestern University, Evanston, IL, United States of America,
**2** Department of Biomedical Engineering, Michigan Technological University, Houghton, MI, United States of America

¤ Current address: Translational and Biomedical Research Center, Joint Department of Biomedical Engineering, Medical College of Wisconsin & Marquette University, Milwaukee, WI, United States of America
* sliu@northwestern.edu

**Data Availability Statement:** All relevant data are within the manuscript and its Supporting Information files.

**Funding:** Funded by NSF.

## Abstract

Ischemic stroke causes acute brain calcium phosphate (CaP) deposition, a process involving primarily the injured neurons. Whereas the adverse impact of CaP deposition on the brain structure and function has been recognized, the underlying mechanisms remain poorly understood. This investigation demonstrated that the neuron-expressed, plasma membrane-associated $Ca^{2+}$-binding proteins annexin (Anx) A2, AnxA5, AnxA6, and AnxA7 contributed to neuronal CaP deposition in the mouse model of ischemic stroke. These Anxs were released from the degraded plasma membrane of the ischemic neurons and were able to form Anx/CaP complexes, a nanostructure capable of binding to the β actin filaments via Anx–actin interaction to cause neuronal CaP deposition prior to brain infarction. Anx administration to the healthy mouse brain caused brain CaP deposition and infarction. Monomeric β actin was able to block competitively Anx binding to β actin filaments and prevent ischemic stroke- and Anx administration-induced brain CaP deposition and infarction. Administration of siRNAs specific to the four Anx mRNAs alleviated brain CaP deposition and infarction. These observations support the role of Anxs in CaP formation and deposition in ischemic neurons.

## Introduction

Ischemic stroke causes brain calcium phosphate (CaP) deposition, a process involving primarily the injured neurons [1–5]. Brain CaP deposition occurs during the early phase of ischemic stroke, as observed in experimental models, and has been considered a process that disrupts neuronal structures, interferes with neuronal functions, and causes neuronal death [3, 5]. However, the molecular mechanisms of ischemic brain CaP deposition remain poorly understood. This investigation demonstrated that several neuron-expressed annexins, including annexin (Anx) A2, AnxA5, AnxA6, and AnxA7, play a role in the development of acute neuronal CaP formation and deposition in the mouse model of ischemic stroke.

Annexins (Anxs) are a family of curve disc-shaped $Ca^{2+}$-binding proteins consisting of 12 members in vertebrates, including AnxA1–AnxA11 and AnxA13 [6–10]. These proteins

**Competing interests:** The authors have declared that no competing interests exist.

primarily attach to the inner cell membrane and have been known to regulate $Ca^{2+}$ trafficking [8], organize cell membrane domains [8], mediate cell membrane repairing processes [11, 12], enhance mineral nucleation in phospholipid-CaP complexes [13], and control $Ca^{2+}$ transport in matrix vesicles for collagen mineralization [14]. An Anx molecule possesses distinct functions on its convex and concave sides–the convex side binds $Ca^{2+}$ [8, 15, 16], whereas the concave side binds protein substrates via its N- or C-terminus [8, 17]. All Anxs possess similar functions, but the number of the bound calcium ions varies [16, 18]. Among the Anx substrates are monomeric and filamentous actin, which exhibits a high binding affinity to Anxs in the presence of $Ca^{2+}$ [8, 17, 19, 20]. The $Ca^{2+}$- and actin-binding functions of Anxs are the molecular basis for CaP formation and deposition to the β actin filaments of the ischemic neurons.

An ischemic condition can cause cell membrane damage [21, 22], resulting in the release of Anxs from the cell membrane and elevation of the cytosolic free Anx level. Cell membrane damage can also cause $Ca^{2+}$ influx from the extracellular space to the cytosol, causing a rapid elevation in the cytosolic $Ca^{2+}$ level [22, 23]. Elevation in both cytosolic Anxs and $Ca^{2+}$ promotes $Ca^{2+}$ binding to Anxs as well as Anx binding to the β actin filaments. As $Ca^{2+}$ binding to Anxs causes focal $Ca^{2+}$ accumulation [16, 18], a condition attracting $PO_4^{3-}$, it can be hypothesized that the cytosolic free Anxs can form Anx/CaP complexes that may in turn deposit to the β actin filaments of the ischemic neurons via Anx–actin interaction. This investigation was designed to test this hypothesis by using the mouse model of ischemic stroke.

## Materials and methods

### Ischemic stroke model

Male, 2 month old, wildtype mice (129S4/SvJaeJ, Jackson Laboratory) were used to establish ischemic stroke. The mice were anesthetized by intraperitoneal injection of Ketamine (100 mg/kg) and Xylazine (10 mg/kg). Ischemic stroke was induced by ligating the right middle cerebral artery and bilateral common carotid arteries for 90 min as described [24]. Note that the ligation of a single middle cerebral artery alone did not cause substantial brain infarction, and the concurrent ligation of a middle cerebra artery and both common carotid arteries enhances the level of brain infarction. Gender- and body weight-matched mice with sham operation were used as controls. Animal distress was alleviated by intramuscular injection of Buprenorphine (0.5 mg/kg). The animals were euthanized by Ketamine (200 mg/kg) and Xylazine (20 mg/kg) overdose followed by cervical dislocation, and examined at selected time points based on the time course of brain calcification with a sample size of 6 mice for each time point determined by statistical power analysis. All experimental procedures were approved by the Northwestern University Institutional Animal Care and Use Committee.

### Measurements of brain infarction and calcification

At a selected time following ischemic stroke, the brain was removed from a deeply anesthetized mouse, frozen at –90˚ C for 5 minutes, and cut into slices of ~0.5 mm in thickness. One of every two consecutive brain slices was incubated in 1% Triphenyl tetrazolium chloride (TTC) at 37˚C for 30 min. The fraction of brain infarcts was measured in reference to the brain volume as described [24]. The remaining brain slices were used to evaluate brain calcification by using the Alizarin Red S assay. The fraction of the calcified brain was measured in reference to the brain volume. For measuring cell-level calcification, the brain of a deeply anesthetized mouse was perfusion-fixed with 4% formaldehyde in PBS and sliced into cryo-sections of 5 µm in thickness. Cell calcification was evaluated by the von Kossa assay along with

immunofluorescence microscopy for identification of the neuron. For fluorescence microscopy, the Alizarin Red S assay was not used as it interferes with fluorescence-based observation.

## Forelimb gripping test

The mouse forelimb muscular function was evaluated based on the gripping force [24]. In this test, one of the mouse forelimbs was wrapped with a piece of Scotch tape to prevent the limb from gripping, and the other forelimb was allowed to grip a wire-ring attached to a force transducer. The mouse was tail-pulled slowly away from the force transducer until the mouse gave up. Changes in the forelimb gripping force were recorded continuously during the test via a data acquisition system and used to analyze the muscular gripping strength. The other forelimb was tested alternately with the same procedure.

## Alkaline phosphatase activity

The relative activity of alkaline phosphatase in brain specimens was analyzed by using the para-nitrophenylphosphate (pNPP) colorimetric assay [25]. Intact and infarcted cerebral cortex specimens were collected at 5, 10, 20, 30, and 60 min following the ligation of the rMCA and bCCAs, homogenized, centrifuged to remove cell debris, and analyzed for total proteins in the supernatant. An amount of 10 μg of total proteins from each sample was added to 1 ml solution containing Tris-HCl 0.1M, $MgCl_2$ 50 mM, $ZnCl_2$ 50 mM, and 5 mM pNPP (pH 9.5) and analyzed by spectrometry at A405 nm. As pNPP reaction is a relatively slow process, measurement outcomes were recorded for each sample at multiple time points, including 5, 10, 20, 30, 40, 50, 60, 90, and 120 minutes, and plotted for comparisons between intact and infarcted cerebral cortex specimens.

## Analysis of Anx expression and distribution in neurons

The relative expression levels of AnxA1 –AnxA11 and AnxA13 were tested in ischemic and sham control neurons isolated from the ischemic and sham control cerebral cortexes by papain (2 mg/ml) perfusion treatment (30˚ C, 30 min) and density-gradient centrifugation of the collected cell suspensions as described [26]. The isolated neurons were treated with a cell-lysing buffer and tested for Anx expression by immunoprecipitation and immunoblotting analyses. β actin from the same sample was used as a loading control. Anx types expressed in the neuron were selected for further analyses.

To test Anx distribution, the cell membrane and cytoskeletal protein fractions of neurons were isolated by using the Qiagen Plasma Membrane Protein kit and the Millipore ProteoExtract® Cytoskeleton Isolation kit, respectively, based on the manufacturers' instructions. The relative levels of the neuron-expressed Anxs, including AnxA2, AnxA5, AnxA6, and AnxA7, in the cell membrane and cytoskeletal fractions were tested by immunoprecipitation and immunoblot analyses and compared between sham control and ischemic stroke. Following Anx preparation, the cytoskeletal fraction was treated with an actin depolymerization buffer (0.1 M PIPES, pH 6.9, 1 mM MgSO4, 10 mM CaCl2, and 5 μM cytochalasin D), and monomeric β actin from the supernatant of the depolymerization buffer-treated cytoskeletal fraction was used as a loading control for immunoblot analysis. Antibodies for Anxs and β actin were from Santa Cruz Biotechnology.

## Visualization of Anxs, actin filaments, and CaP deposits

The distribution of Anxs and β actin filaments was tested by fluorescence microscopy, and CaP deposition was detected by the von Kossa assay. Brain specimens were perfusion-fixed by

using 4% formaldehyde in PBS, cut into cryo-sections of 5 μm in thickness, incubated with an anti-AnxA2, AnxA5, AnxA6, or AnxA7 antibody and subsequently with FITC-conjugated phalloidin (1 μM), treated with a von Kossa solution, and examined by fluorescence and bright-field microscopy. Hoechst 33258 was used to label cell nuclei. The same specimen was used for both fluorescence and bright-field microscopic analyses.

To demonstrate Anx translocation to the β actin filament compartment, the fraction of Anx overlap with β actin filaments was measured by fluorescence microscopy based on the different fluorescence colors on an anti-Anx (A2, A5, A6, or A7) antibody (red) and the FITC-phalloidin-labeled β actin filaments (green). The yellow color represents overlap of the red-colored Anx with the green-colored β actin filaments in a fluorescence image. Brain specimens were prepared from 6 mice with 5-day sham operation and 6 mice with 5-day ischemic stroke, fixed in 4% formaldehyde/PBS, and cut into 5 μm sections. From each specimen section, six randomly selected regions were used for analyses. The areas of each red-colored Anx (A2, A5, A6, or A7), the green-colored β actin filaments, and the yellow-colored Anx–β actin filament overlap regions were measure by using the NIH Image J image processing system for the calculation of the fraction of Anx overlap with the β actin filaments.

## Analysis of Anx binding to monomeric β actin

Anx binding to monomeric β actin was evaluated by immunoprecipitation and immunoblot analyses. Neurons were isolated from the ischemic and sham control cerebral cortexes as above, treated with a cell-lysing buffer, and used for immunoprecipitation with an anti-β actin antibody (Santa Cruz Biotechnology) as described [27]. Immunoblot analysis was carried out by using an anti-AnxA2, AnxA5, AnxA6, or AnxA7 antibody (Santa Cruz Biotechnology).

## Co-sedimentation of β actin filaments with Anxs

Anx binding to β actin filaments was tested by using the actin filament co-sedimentation assay and immunoblot analysis [28, 29]. β actin was isolated from the cerebral cortex neurons by immunoprecipitation, protein denaturation, SDS-polyacrylamide gel electrophoresis, elution in 0.1% SDS solution, acetone precipitation, and renaturation [30–32]. The presence of β actin was confirmed by immunoblot analysis. The isolated β actin (~5 μM) was allowed to polymerize in a buffer containing 50 mM KCl, 2 mM MgCl$_2$, 5 mM Tris-Cl (pH 8.0), and 1 mM ATP at 23˚ C for 30 min [29]. AnxA2, AnxA5, AnxA6, and AnxA7 (~5 μM) were isolated from cortical neurons by using the same method above. Each Anx was introduced to polymerized β actin filaments in the same buffer with the addition of 1 mM calcium chloride and incubated at 37˚ C for 60 min. Control sedimentation was performed in the absence of an Anx. The Anx/actin filament mixture was placed over a 50 μl cushion of 30% sucrose and centrifuged at 14,000 g for 60 min [33]. The β actin filament pellet was washed, collected, denatured, and centrifuged. The supernatant of the denatured actin filaments was used to test the relative level of each Anx by immunoblot analysis. The presence of Anxs in the β actin filament fraction indicates Anx binding to β actin filaments. Following Anx preparation, the β actin filament fraction was treated with an actin depolymerization buffer (0.1 M PIPES, pH 6.9, 1 mM MgSO$_4$, 10 mM CaCl$_2$, and 5 μM cytochalasin D), and monomeric β actin from the supernatant of the treated β actin filaments was used as a loading control for immunoblot analysis.

## Analysis of Anx-mediated CaP formation

To test whether Anxs cause CaP formation, AnxA2, AnxA5, AnxA6, or AnxA7 (400 ng/ml) derived from neurons was mixed in PBS (PO$_4^{3-}$ 4 mM) with 1 mM CaCl$_2$, incubated at 37˚C for 60 minutes, and examined for the formation of CaP nanoparticles by optical microscopy

and atomic force microscopy (MFP3D, Asylum) at a scan rate of 0.3 Hz and scan lines of 512. PBS ($PO_4^{3-}$ 4 mM) with 1 mM $CaCl_2$ without Anxs was used as a control. In addition, intact and ischemic cerebral cortex specimens were prepared from mice with 5-day ischemic stroke, fixed in 4% formaldehyde, cut into cryo-sections of 5 μm in thickness, and examined by optical microscopy and atomic force microscopy for CaP formation and deposition. Specimen sections from the mouse femur were prepared and used as an apatite standard for comparison.

### Evaluation of Anx/CaP aggregate formation on brain specimens in situ

Freshly prepared brain specimens were cut into cryo-sections of 10 μm in thickness without chemical fixation. AnxA2, AnxA5, AnxA6, or AnxA7 (400 ng/ml in PBS with $PO_4^{3-}$ 4 mM and $CaCl_2$ 1 mM) was used to treat the brain specimen sections at 37°C for 1 hour in the presence or absence of recombinant monomeric β actin (1 μg/ml), or phalloidin (200 nM). PBS with $PO_4^{3-}$ 4 mM and $CaCl_2$ 1 mM was used as a control. Specimens were treated with Alizarin Red S and used for measuring the area fraction of CaP aggregates in reference to the total area observed by optical microscopy.

### Administration of Anx siRNAs and recombinant Anxs

To test the role of Anxs in brain CaP deposition *in vivo*, an AnxA2, AnxA5, AnxA6, or AnxA7 siRNA or a combination of all 5 siRNAs (5 μg/ml each in a transfection medium with 2% liposome-based siRNA transfection reagent from Santa Cruz Biotechnology) was used to impregnate a polyethylene fiber patch (0.5 mm x 4 mm x 4 mm). A scrambled siRNA was used as a control. The polyethylene fiber patch was placed onto the brain surface after removing the dura mater by craniotomy and covered on the non-brain contact side with a thin latex membrane to prevent siRNA loss. Ischemic stroke was induced at 3 days after siRNA administration, a period allowing effective silencing of a target gene. Anx expression, forelimb gripping strength, brain CaP deposition, and brain infarction were tested following ischemic stroke and compared between Anx and control siRNA administrations.

The pro-CaP deposition action of Anxs was also tested by Anx protein administration. A recombinant AnxA2, AnxA5, AnxA6, or AnxA7 protein (50 ng each in 10 μl PBS with $PO_4^{3-}$ 4 mM) was delivered to the brain by using the polyethylene fiber patch method as above. PBS with $PO_4^{3-}$ 4 mM was used as a control. Brain CaP deposition was assessed by the Alizarin Red S assay at 5 days following administration. Results were analyzed and compared between Anx and PBS administrations.

### Statistics

Means and standard deviations were calculated for all measured parameters. The Student's t-test was used for comparisons between two groups. The analysis of variance (ANOVA) test was used for comparisons between multiple groups. Post-hoc comparisons were carried out by the Bonferroni method. Sample sizes were estimated by power analysis at the power level of 0.8 and α level of 0.05. A difference was considered statistically significant at $p < 0.05$.

## Results

### Characterization of ischemic brain CaP deposition

Ischemic stroke caused rapid brain CaP deposition in primarily the ischemic neurons. Brain CaP deposition was found in the ischemic cortex region as early as 90 minutes following the ligation of the right middle cerebral artery (rMCA) and bilateral common carotid arteries (bCCAs) with a progressive increase in the CaP level as observed by the Alizarin Red S assay

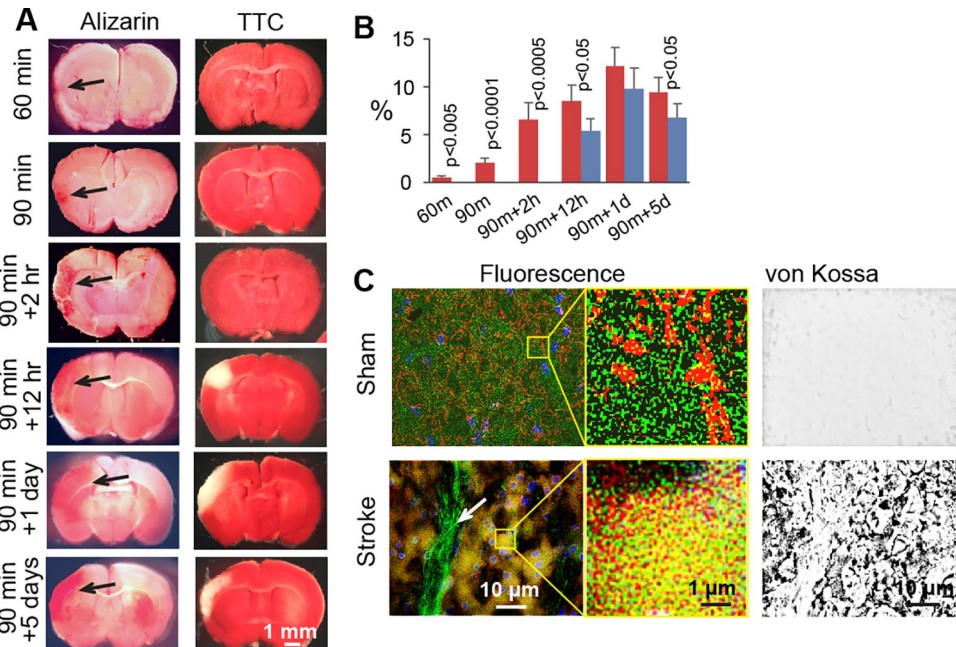

**Fig 1. Characterization of brain CaP deposition in ischemic stroke.** A. Bright-field micrographs showing brain CaP deposition (red by the Alizarin Red S assay) and brain infarction (pale by the TTC assay) in the area of the right premotor/primary motor cortex following ischemic stroke. At each time point, two adjacent brain slices (0.5 mm in thickness) were prepared from the center of the ischemic region for Alizarin Red S and TTC assays (one slice for each assay). Arrows: Sites of CaP deposition. 60 min and 90 min are durations of ligation of the right middle cerebral artery and bilateral carotid arteries; and +2 hr, +12 hr, +1 day, and +5 days are durations of blood flow reperfusion following cerebral artery ligation. B. Graphic representation of the fractions of the CaP-positive (red) and infarcted (blue) brain. m: Minutes. h: Hours. d: Days. Means and SDs are presented (n = 6). ANOVA p < 0.0001 between brain CaP deposition and infarction for all time points. The p value on each selected bar is for comparison between brain CaP deposition and infarction at a time point. C. Immunofluorescence and bright-field (von Kossa) micrographs showing the distribution of the neuron-specific protein microtubule-associated protein 2 (MAP2, red), actin filaments (green), and CaP deposition (black in von Kossa images) in sham control and ischemic brain specimens at 5 days. Blue: Cell nuclei. Yellow: Overlap of red and green. Arrow: Actin filaments in vascular cells.

(Fig 1A). An important finding was that the Alizarin Red S-detected ischemic brain CaP deposition occurred earlier than the TTC-detected brain infarction, a process first found at 12 hours of blood flow reperfusion following 90 min ligation of the rMCA and bCCAs (Fig 1A). These observations suggest that ischemic brain CaP deposition possibly contributed to brain injury as CaP crystals can disrupt cell structures and interfere with cell functions. In the ischemic region, the CaP-positive area was larger than that of the infarcted brain at a given time (p < 0.0001 over all observation time points by ANOVA, Fig 1B), suggesting that brain CaP deposition is more susceptible to ischemia than brain infarction. Furthermore, CaP deposition was found in the ischemic, but not in the intact neurons, which were identified based on the expression of neuron-specific microtubule-associated protein 2 (MAP2). An interesting observation was that MAP2 was present in the form of non-uniformly distributed aggregates in the intact neurons, but was dispersed into more uniformly distributed smaller particles through the ischemic neurons (Fig 1C). In association with brain CaP deposition and infarction, ischemic stroke caused a significant decrease in the forelimb gripping strength. The left forelimb exhibited a more significant impairment in gripping strength than the right forelimb of the same mouse with right middle cerebral artery ligation (Fig 2).

Another early change in the ischemic brain was the activation of alkaline phosphatase, an enzyme catalyzing the generation of the CaP element $PO_4^{3-}$ from inorganic pyrophosphate

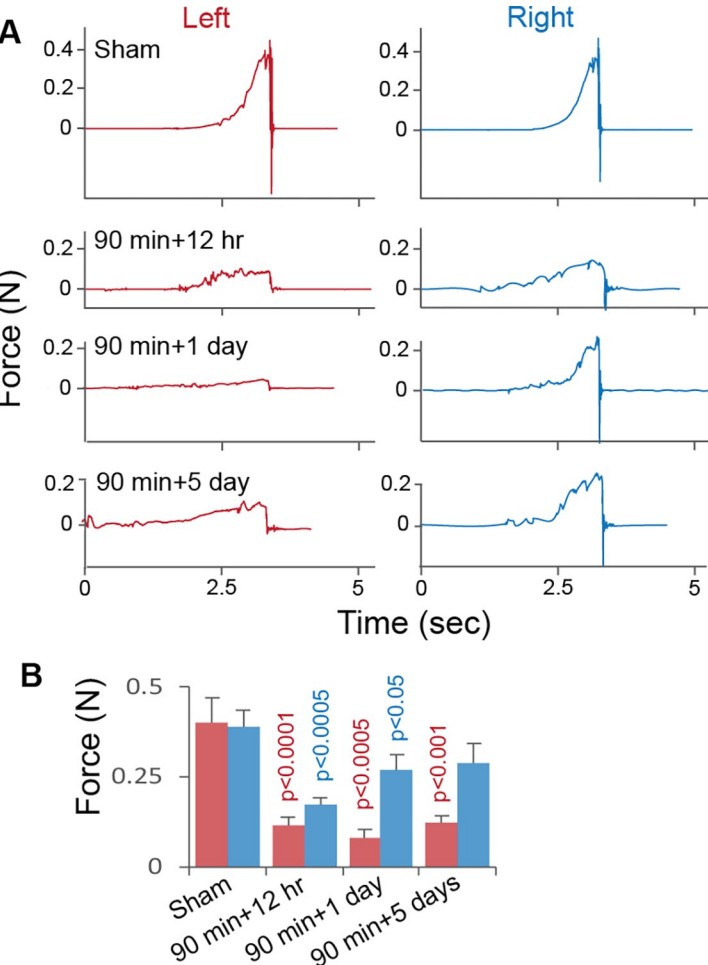

**Fig 2. Forelimb-gripping forces in mice with sham operation and ischemic stroke.** A. Original experimental records of the mouse left (red) and right (blue) forelimb-gripping forces in sham control and ischemic stroke. N: Newton. 90 min is the duration of the ligation of the right middle cerebral artery and bilateral common carotid arteries. +12 hr, +1 and +5 days are durations of blood flow reperfusion. B. Graphic representation of mouse forelimb forces. Means and SDs are presented (n = 6). The red and blue bars represent the left and right forelimb forces, respectively. The red- and blue-colored p values are for comparisons in the left and right forelimb forces, respectively, between ischemic stroke and sham control at each time point.

[34], as detected by the p-nitrophenyl phosphate (pNPP) colorimetric assay (Fig 3). This test showed a progressive increase in the alkaline phosphatase activity with initial detection at 5 minutes following right middle cerebral artery and bilateral common carotid artery ligation. As an increase in the $PO_4^{3-}$ level promotes CaP formation [25], early activation of alkaline phosphatase, together with ischemia-induced elevation of cytosolic $Ca^{2+}$, may facilitate CaP formation and deposition in the ischemic neurons.

## Anx expression and distribution in sham control and ischemic neurons

The mouse cortical neurons were found to express AnxA2, AnxA5, AnxA6, and AnxA7 as tested by immunoblot analysis (Fig 4A). Other Anx family proteins, including AnxA1, AnxA3, AnxA4, AnxA8, AnxA9, AnxA10, AnxA11, and AnxA13, were expressed at much lower levels (Fig 4A) and were not tested further in this investigation. Ischemic stroke did not cause a substantial change in the relative expression of AnxA2, AnxA5, AnxA6, and AnxA7 in the cortical

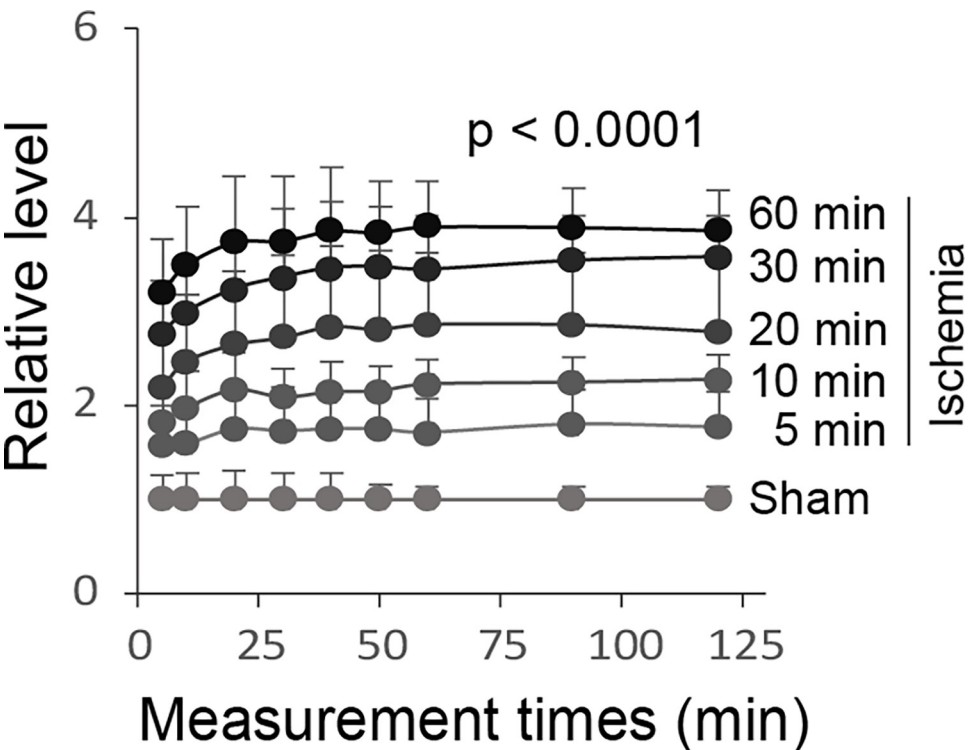

**Fig 3. Activation of alkaline phosphatase in the cortex in response to ischemic stroke.** The figure shows the relative levels of alkaline phosphatase activity in homogenized sham control and ischemic cortex specimens detected by using the p-nitrophenyl phosphate (pNPP) assay at spectrometric absorbance wavelength 405 nm. Means and SDs are presented (n = 8). ANOVA p < 0.0001 across sham-control and ischemic cortex specimens at 5, 10, 20, 30, and 60 minutes. All measured values were normalized in reference to the mean of the sham-control level at each spectrometry measurement time. The x axis represents time points at which specimens were measured by spectrometry.

neurons in reference to sham controls (Fig 4A). However, there was a substantial change in the distribution of the neuron-expressed Anxs in response to ischemic stroke. Anxs were primarily present in the plasma membrane fraction of sham control neurons, whereas higher

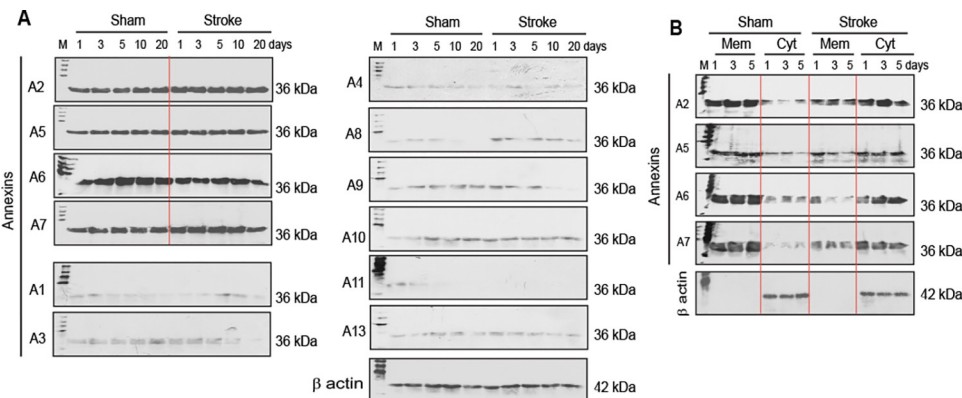

**Fig 4. Anx expression and distribution in sham control and ischemic neurons.** A. Immunoblot analysis of Anx expression in sham control and ischemic neurons. M: Molecular weights. B. Immunoblot analysis of Anx distribution in the cell membrane and cytoskeletal fractions of sham control and ischemic neurons. Mem: Cell membrane fraction. Cyt: Cytoskeletal fraction.

levels of Anxs were detected in the cytoskeletal fraction compared with the levels in the cell membrane fraction of ischemic neurons (Fig 4B). These observations suggest that the neuron-expressed Anxs translocate from the cell membrane to the cytoskeleton of ischemic neurons.

## Anx binding to monomeric and filamentous β actin

Co-immunoprecipitation and immunoblot analyses showed that the neuron-expressed Anxs (AnxA2, AnxA5, AnxA6, and AnxA7) were able to bind to monomeric β actin (Fig 5A). A co-sedimentation assay demonstrated that these Anxs were able to bind to β actin filaments *in vitro* (Fig 5B). Pretreatment of Anxs with monomeric β actin substantially prevented Anx binding to β actin filaments (Fig 5B), suggesting that monomeric β actin was able to competitively bind to the Anx sites responsible for interacting with β actin filaments. Stabilization of β actin filaments with phalloidin reduced the affinity of Anx binding to β actin filaments *in vitro*

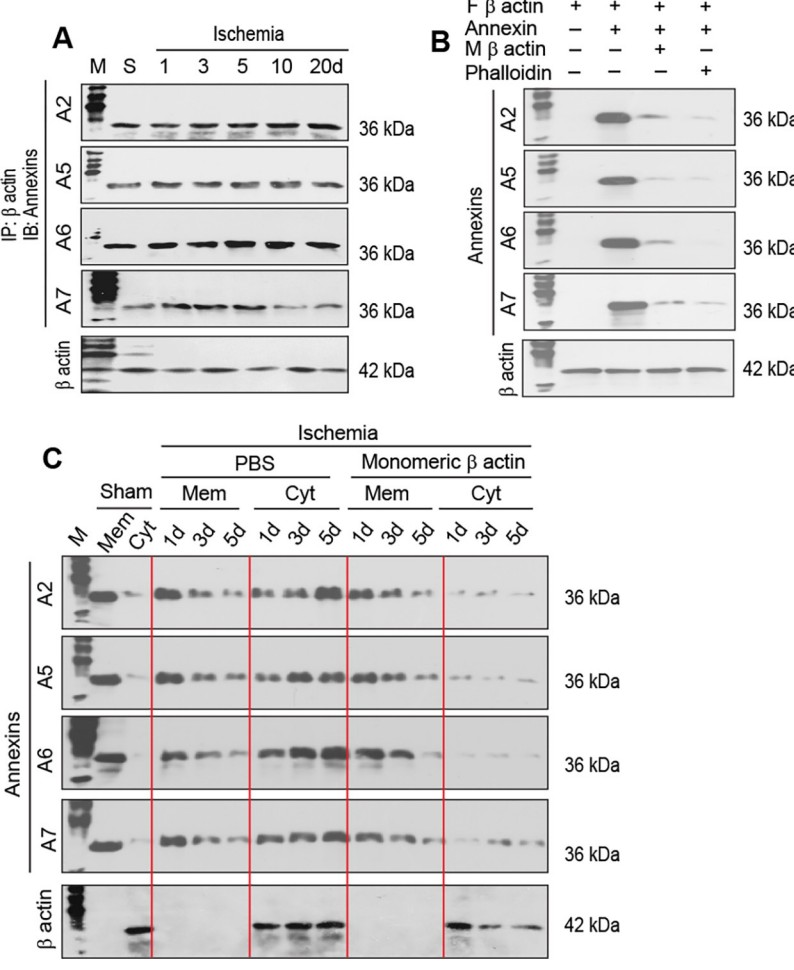

**Fig 5. Anx binding to monomeric and filamentous β actin.** A. Co-immunoprecipitation (IP) and immunoblot (IB) analyses of Anxs and β actin from sham control (S) and ischemic neurons. The sham control neurons were prepared at 5 days following sham operation. B. Co-sedimentation of filamentous (F) β actin with Anxs in the presence and absence of monomeric (M) β actin or phalloidin in vitro. C. Immunoblot analysis of Anx translocation from the cell membrane to the cytoskeletal fraction of ischemic neurons in the presence and absence of monomeric β actin. Mem: Cell membrane fraction. Cyt: Cytoskeletal fraction. In panels B and C, control β actin was derived from the filamentous β actin fraction (panel B) and cytoskeletal fraction (panel C) depolymerized by using a cytochalasin D-containing F-actin depolymerization buffer.

(Fig 5B). Administration of monomeric β actin to the ischemic brain immediately following the induction of ischemic stroke prevented Anx translocation from the cell membrane to the cytoskeletal fraction of the cortical neurons (Fig 5C). These observations suggest that the β actin filaments are a cytoskeletal binding target for the neuron-expressed Anxs, and monomeric β actin can be used to block Anx binding to the target.

## Role of Anxs in brain CaP deposition

Anxs are capable of binding and carrying $Ca^{2+}$ with the ion number per molecule depending on the Anx type [6, 8, 15]. An AnxA2, AnxA5, or AnxA6 molecule can bind up to 7 [18], 10 [16], or 5 [34] calcium ions, respectively (note that information for AnxA7 is not available). These calcium ions mediate Anx attachment to the inner cell membrane. In ischemic stroke, the neuronal plasma membrane undergo disintegration [21, 22], a process potentially freeing Anxs into the cytosol. The cytosolic $Ca^{2+}$ and $PO_4^{3-}$ levels are increased because of elevated $Ca^{2+}$ influx [22, 23] and activation of alkaline phosphatase (Fig 3 in this report), respectively. These changes may promote $Ca^{2+}$ loading onto the free Anx molecules, cause $Ca^{2+}$ accumulation on Anxs, and thereby facilitate $PO_4^{3-}$ binding to the Anx-carried calcium ions to form Anx/CaP nano-complexes, which in turn deposit to β actin filaments via Anx–β actin interaction in ischemic neurons.

The aforementioned mechanisms of Anx/CaP deposition are supported by several observations. As shown by fluorescence microscopy, the sham-control mice did not exhibit substantial overlaps of AnxA2, AnxA5, AnxA6, or AnxA7 with β actin filaments in cortical neurons with undetectable CaP deposition. In contrast, the mice with ischemic stroke showed a significant increase in the fraction of overlap of all tested Anxs with β actin filaments in association with CaP deposition (Fig 6A and 6B). In the presence of $Ca^{2+}$ (1 mM) and $PO_4^{3-}$ (4 mM) *in vitro*, Anxs were able to assemble $Ca^{2+}$ and $PO_4^{3-}$ into bone apatite-like nanoclusters, analogous to those found in the CaP-positive ischemic cortex (Fig 7). Each Anx (AnxA2, AnxA5, AnxA6, or AnxA7) was able to form μm-sized Anx/CaP aggregates in the presence of $Ca^{2+}$ (1 mM) and $PO_4^{3-}$ (4 mM) *in vitro* as tested by the Alizarin red S assay (Fig 8). These aggregates were able to bind to freshly prepared, non-fixed healthy brain specimen sections *in vitro* (Fig 8). Pretreatment of Anx/CaP aggregates with monomeric β actin significantly prevented Anx/CaP aggregate binding to the brain specimen sections (Fig 8). Pretreatment of the brain specimen sections with phalloidin exerted a similar inhibitory effect (Fig 8), suggesting that phalloidin might cause a change in β actin arrangement within an actin filament, which prevents Anx access to the binding sits of actin molecules. Results from these tests provided additional evidence supporting the role of Anxs in the formation of Anx/CaP aggregates, interaction between Anxs and β actin filaments, and deposition of Anx/CaP aggregates to the β actin filaments. It should be noted that healthy brain specimens, instead of ischemic brain specimens, were used in these tests because ischemic stroke caused annexin–β actin filament interaction, making it difficult to distinguish between the *in vitro* induced annexin–β actin filament interaction and the ischemia-induced annexin–β actin filament interaction.

Further tests showed that administration of AnxA2, AnxA5, AnxA6, or AnxA7 to the healthy mouse brain caused brain CaP deposition, as tested by the Alizarin Red S assay, in association with brain infarction, as detected by the TTC assay (Fig 9A and 9B). Co-administration of an Anx type with monomeric β actin to the healthy brain significantly prevented Anx-induced brain CaP deposition and infarction (Fig 9A and 9B). Whereas Anxs were detected in the cytoskeletal fraction of cortical neurons in response to Anx administration, co-administration with monomeric β actin prevented Anx binding to the cytoskeleton (Fig 9C). Administration of each Anx type to the right hemisphere of the healthy brain caused a

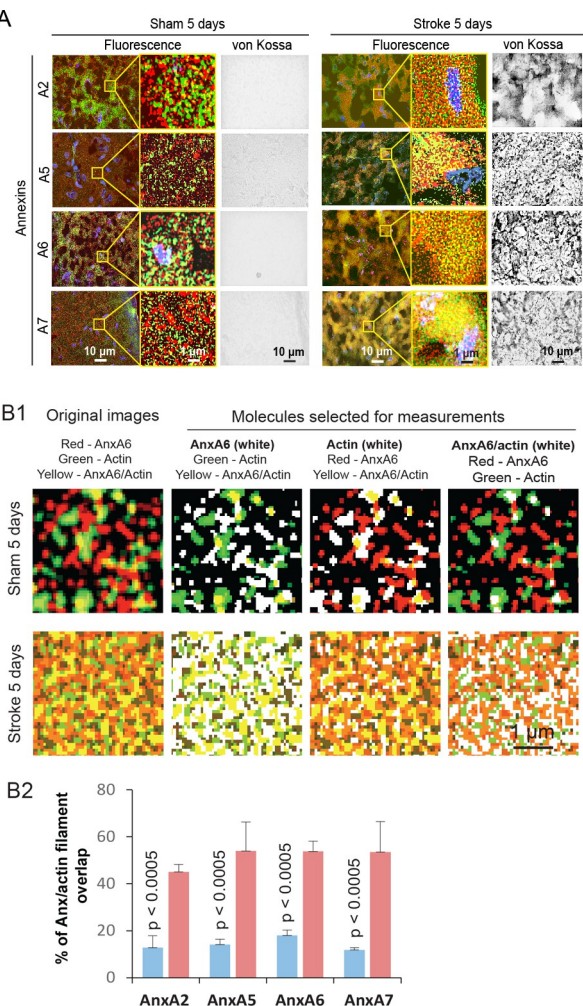

**Fig 6. Distribution of Anx and β actin filaments in the cortical neurons of mice with sham operation and ischemic stroke.** (A) Fluorescence and bright-field (von Kossa) micrographs showing the expression and distribution of Anxs (red), actin (green), and CaP deposition (black in von Kossa images) in sham control and ischemic cortical specimens prepared at 5 days. The von Kossa and fluorescence images for each Anx were prepared from the same site of each sham-control or ischemic specimen. Each scale bar is for all images of each column. (B) Distribution of AnxA2, AnxA5, AnxA6, and AnxA7 in reference to the distribution of β actin filaments in the cortical neurons of mice with sham operation and ischemic stroke at 5 days. (B1) Fluorescence images showing fluorescent color-based selection of AnxA6, as an example (originally labeled red), β actin filaments (originally labeled green), and the regions of AnxA6 overlap with β actin filaments (originally yellow) by using the NIH ImageJ image processing system. The white color represents a selected molecule. (B2) Graphic representation of the fraction of Anx overlap with β actin filaments. Means and SDs are presented (n = 6 for each Anx). An increase in Anx–β actin filament overlap is indicative of Anx translocation to the β actin filament compartment.

significant decrease in the left forelimb gripping strength, whereas co-administration with monomeric β actin significantly alleviated Anx-induced impairment of forelimb gripping strength except for that by AnxA6 (Fig 10).

The siRNA-mediated Anx gene silencing approach was used to confirm the role of Anxs in ischemic brain CaP deposition. siRNA was able to diffuse into the brain (Fig 11A) when delivered by using a polymer-patch method described in the Methods section. Administration of AnxA2, AnxA5, AnxA6, or AnxA7 siRNA substantially reduced the relative expression level of each Anx at 3 and 6 days (Fig 11B). In ischemic stroke induced at 3 days after Anx siRNA

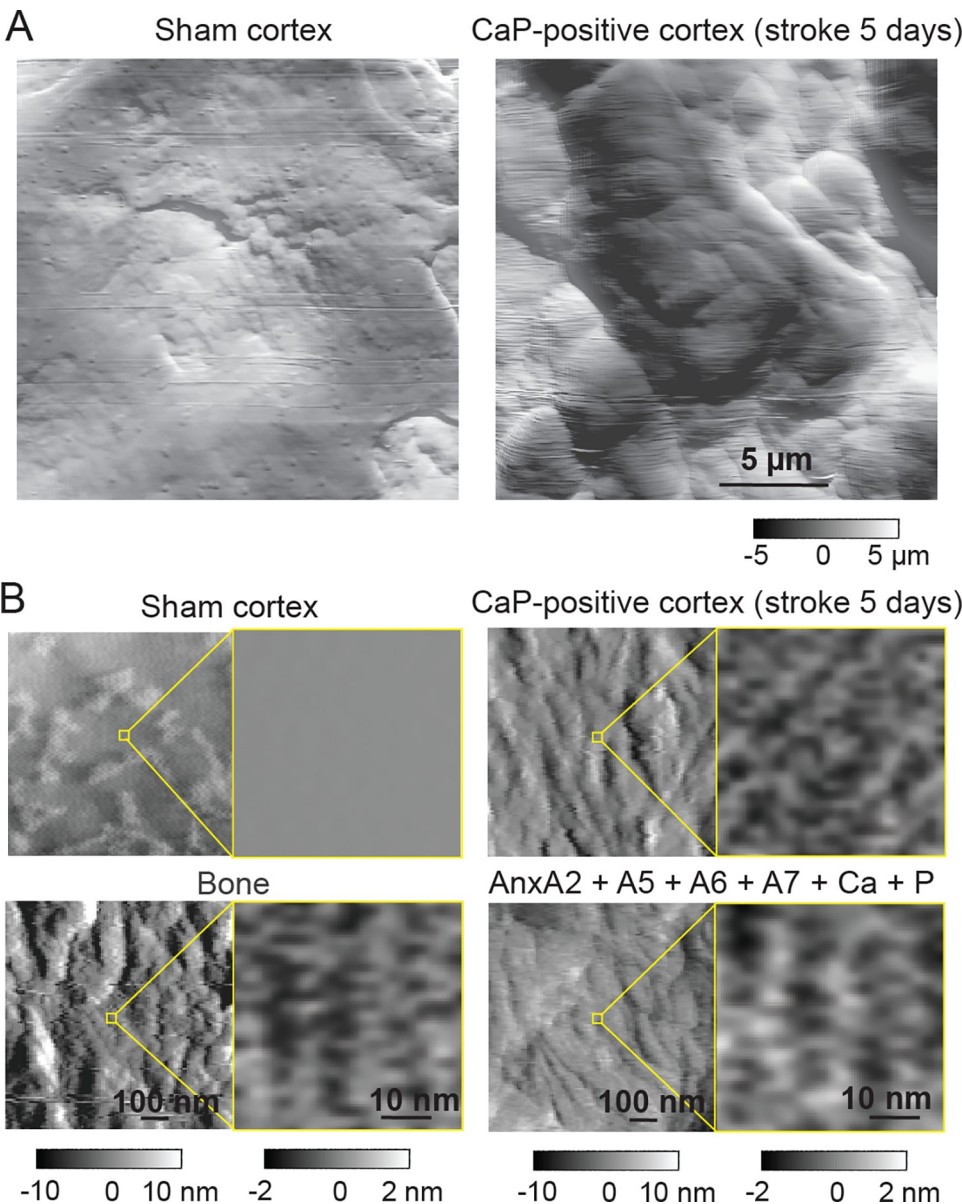

**Fig 7. Nanostructures of mouse sham CaP-negative and ischemic CaP-positive cortices, mouse femur, and in vitro constructed Anx/CaP aggregates visualized by atomic force microscopy (AFM).** A. μm-level AFM structures of the sham and ischemic cortical regions at 5 days. The length scale is for both panels. The black-to-white gradient bar indicates the structure height for both panels. B. Comparisons in nanostructure between the sham CaP-negative cortex, ischemic CaP-positive cortex (5-day stroke), femur, and constructed Anx/CaP aggregates. Ca: $Ca^{2+}$. P: $PO_4^{3-}$. Each length scale is for both panels of each column. Each black-to-white gradient bar indicates the height of nanostructures for both panels of each column.

administration, the levels of brain CaP deposition and infarction were significantly reduced in response to Anx gene silencing (except for AnxA2 siRNA) in reference to the levels with control siRNA administration (Fig 11C and 11D). Administration of AnxA2, AnxA5, AnxA6, and AnxA7 siRNAs in combination further reduced the levels of brain CaP deposition and infarction (Fig 11C and 11D). Whereas administration of a single Anx siRNA did not significantly improve the gripping strength of the impaired forelimb, administration of AnxA2, AnxA5,

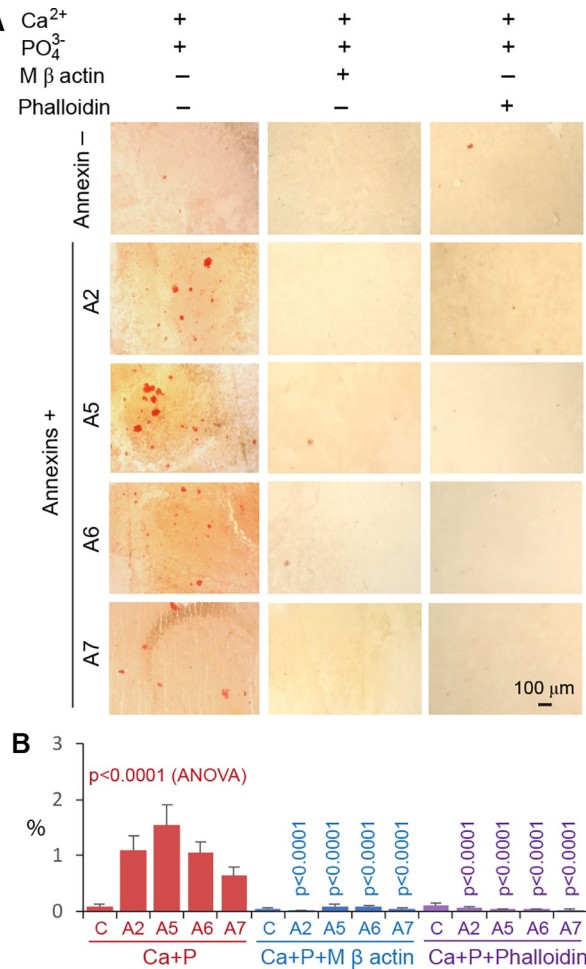

**Fig 8. Formation and attachment of Anx/CaP aggregates to freshly prepared, non-fixed cortex sections in situ.** A. Bright-field micrographs showing Alizarin Red S-stained Anx/CaP aggregates on cortex sections in the presence (+) or absence (−) of monomeric (M) β actin or phalloidin. The length scale is for all panels. B. Graphic representation of the area fractions of Anx/CaP aggregates on cortex sections in the presence or absence of monomeric β actin or phalloidin. Means and standard deviations are presented (n = 6). The ANOVA p value is for comparison across control (C, without Anx) and various Anx types in the presence of Ca2+ (Ca) and PO43- (P). The vertical p value on each bar is for comparison between groups of each Anx with and without monomeric β actin (blue) or phalloidin (purple). A2, A5, A6, and A7: AnxA2, AnxA5, AnxA6, and AnxA7, respectively.

AnxA6, and AnxA7 siRNAs in combination significantly alleviated the impairment of the fore-limb gripping strength (Fig 12). Taken together, these observations support the contribution of Anxs to brain CaP deposition and infarction.

## Discussion

CaP deposition occurs in the ischemic brain of the human and is often found from several months to years after ischemic stroke in selected cases [2, 4]. In contrast to these clinical observations, experimental investigations have shown a much earlier occurrence of brain CaP deposition (within several hours) with a higher incidence in the mouse model of ischemic stroke [5]. In the present investigation, all tested animals showed brain CaP deposition within 90 minutes following the induction of ischemic stroke. The discrepancies between the clinical and experimental observations may be attributed to the different methods used for CaP

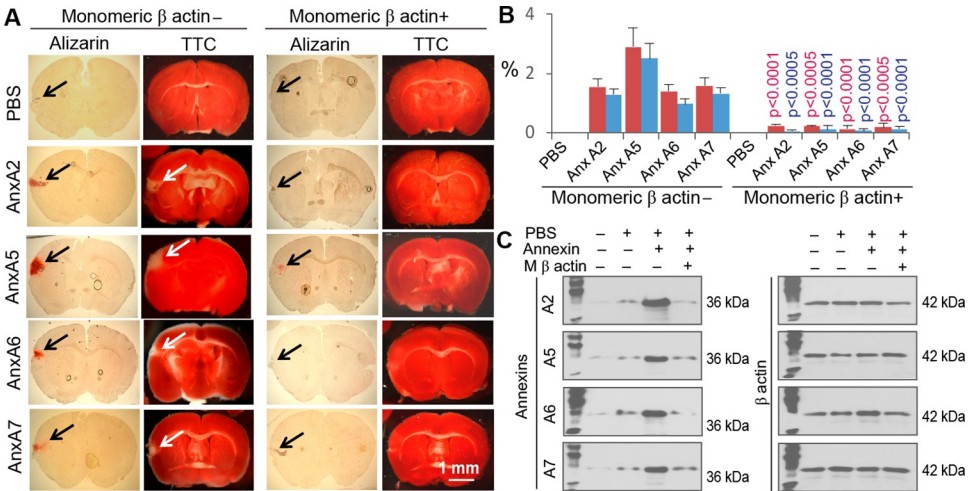

**Fig 9. Brain CaP deposition and infarction in response to Anx administration to the right premotor/primary motor cortex of healthy mice.** A. Brain CaP deposition (red in color by the Alizarin Red S assay) and brain infarction (pale by the TTC assay) in response to administration of an Anx (10 μl, 400 ng/ml in PBS) in the presence or absence of monomeric β actin (1 μg/ml) at 5 days. To prepare specimens for these assays, two fresh brain slices of 1 mm in thickness was prepared at the Anx injection site. One brain slice was cut into cryo-sections of 50 μm in thickness for Alizarin Red S assay, and the other brain slice was used for TTC assay. Black arrows: CaP deposition at Anx administration sites. White arrows: Brain infarcts at administration sites. B. Graphic representation of the area fractions of CaP-positive (red) and infarcted (blue) brain in response to administration of PBS, AnxA2, AnxA5, AnxA6, or AnxA7 in the presence or absence of monomeric β actin at 5 days. Means and SDs are presented (n = 6). Each red- and blue-colored p values are for comparison in the area fractions of the CaP-positive and infarcted brains, respectively, between administrations of each Anx with and without monomeric β actin. C. Anx binding to the cytoskeleton of cortical neurons in response to Anx administration to the healthy mouse brain in the presence or absence of monomeric β actin at 5 days. M β actin: Monomeric β actin.

detection. In the clinical setting, brain CaP deposition is often evaluated by CT and MRI [2, 4], whereas experimental CaP deposition is tested by colorimetric assays based on phosphate chemical reactions, such as the BCIP/NBT, pNPP, Alizarin Red S, and von Kossa assays. CT and MRI may not be sensitive to early mild-level CaP deposition. In contrast, the colorimetric assays can be used to detect low levels of CaP deposition, but it is difficult to use these assays in patients. Thus, experimental modeling of brain CaP deposition provides insights into the pathogenic mechanisms and impact of brain CaP deposition in ischemic stroke.

It is important to evaluate the timing of brain CaP deposition in reference to the occurrence of brain infarction. CaP deposition after neuronal death is not considered a factor contributing to neuronal death, whereas CaP deposition prior to neuronal death potentially contributes to neuronal death as CaP formation and deposition disrupt cytoplasmic structures and interfere with neuronal functions. In the mouse model of ischemic stroke, the peak rate of neuronal death occurs at 24 hours as detected by the TUNEL assay [24]. The present investigation showed that the activity of alkaline phosphatase was elevated in the ischemic cortex at 5 min of stroke, suggesting a rapid generation of $PO_4^{3-}$. This process, together with cytosolic $Ca^{2+}$ overload in the injured neurons, establishes a condition suitable for CaP formation. As shown in this report, the first sign of brain CaP deposition were found at 90 min of ischemic stroke by the Alizarin Red S assay, whereas the first sign of brain infarction was observed at 13.5 hours (90 min of cerebral ischemia + 12 hours of blood flow reperfusion) by the TTC assay, showing that brain CaP deposition occurs about 12 hours earlier than brain infarction. These analyses suggest that brain CaP deposition potentially contributes to brain infarction. This concept is

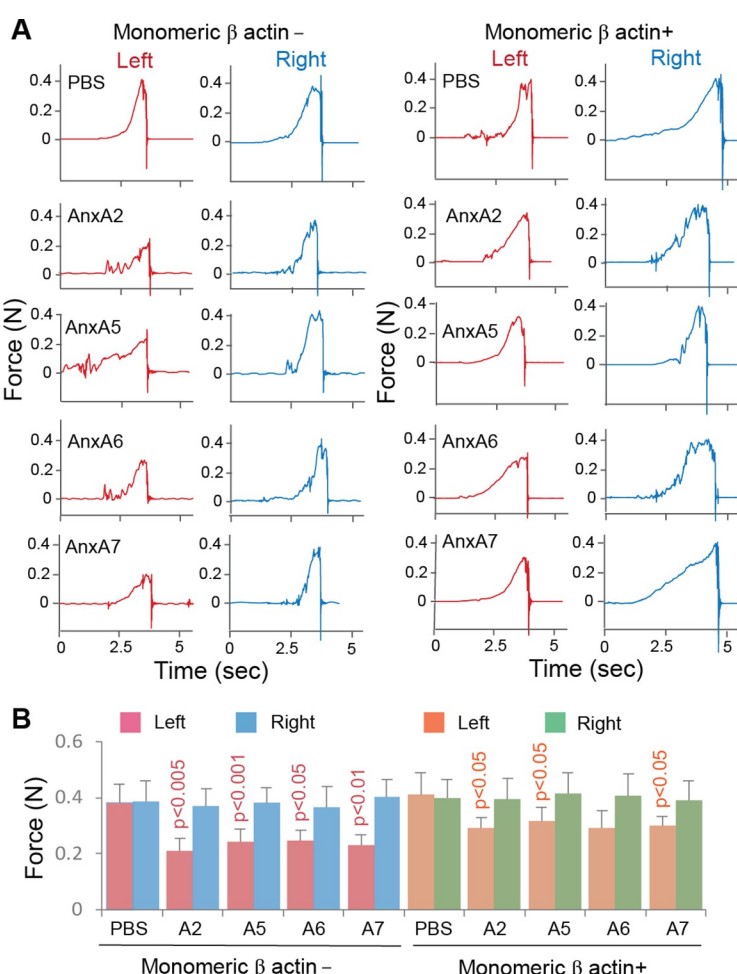

**Fig 10. Changes in forelimb gripping strength in response to Anx administration to the premotor/ primary motor cortical region of healthy mice with and without monomeric β actin.** A. Experimental records of the left and right forelimb forces at 5 days following cortical administration of PBS or an Anx in PBS with and without monomeric β actin. N: Newton. B. Graphic representation of mouse forelimb forces at 5 days following administration of PBS or an Anx with and without monomeric β actin. Means and SDs are presented (n = 6). The p value on a bar of the monomeric β actin-negative graph represents the significance level of difference in the left forelimb strength between the administrations of PBS and an Anx (note that ischemic stroke was induced in the right hemisphere). The p value on a bar of the monomeric β actin-positive graph represents the significance level of difference in the left forelimb strength between the administrations of an Anx with and without monomeric β actin (comparisons between the monomeric β actin-positive and negative graphs).

supported by the observation that Anx administration-induced brain CaP deposition caused brain infarction in the absence of ischemic stroke (Fig 9).

The primary goal of this investigation is to test the mechanisms of neuronal CaP formation and deposition in ischemic stroke. Anx translocation, Anx/CaP nano-complex formation, and Anx/CaP binding to β actin filaments, as demonstrated in this report, are potential processes for neuronal CaP deposition (Fig 13). Anxs are $Ca^{2+}$-binding proteins attached to the inner surface of the cell membrane via the mediation of $Ca^{2+}$ [6, 8, 15]. In ischemic stroke, Anxs may be freed from the damaged cell membrane. The free Anxs can accept $Ca^{2+}$ to cause focal $Ca^{2+}$ accumulation, a condition to attract $PO_4^{3-}$ and form Anx/CaP nano-complexes. As Anxs possess a high binding affinity to actin [8, 17, 19, 20], the Anx/CaP nano-complexes may bind to β actin filaments via Anx-actin interactions, thereby causing Anx/CaP deposition. The relative

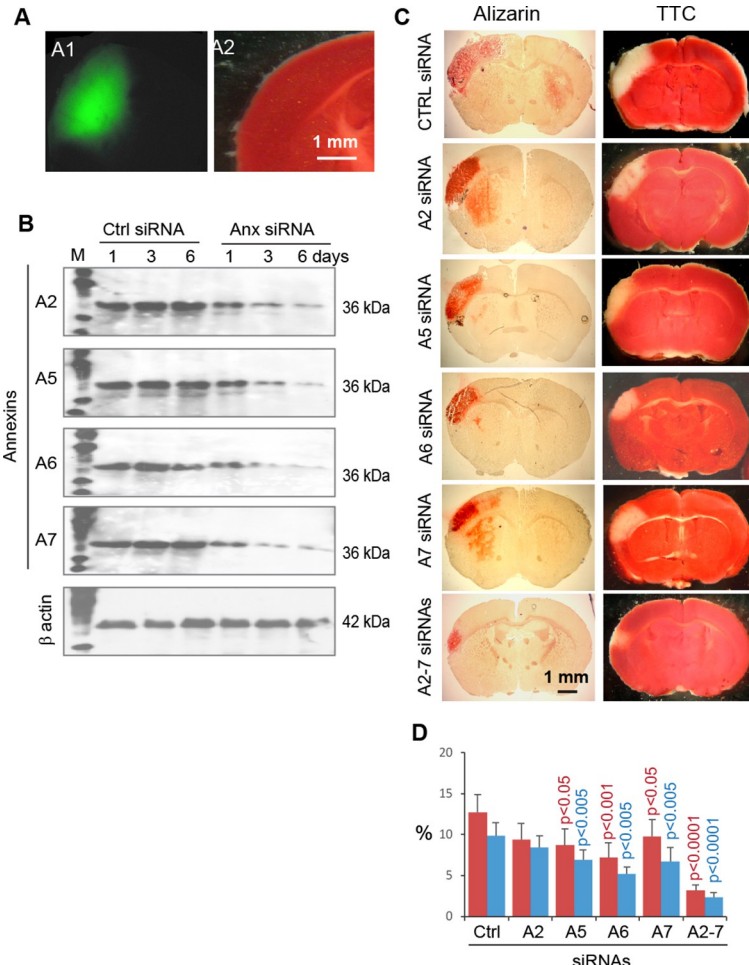

**Fig 11. Alleviation of brain CaP deposition and infarction in response to Anx gene silencing in ischemic stroke.** A. Administration and detection of FITC-conjugated AnxA2 siRNA in the mouse cerebrum. A1. Fluorescence micrograph of a healthy cerebrum at 1 hour following administration of FITC-conjugated AnxA2 siRNA, showing the spreading of the injected siRNA in the cortical and adjacent regions. A2. Bright-field micrograph of the same brain specimen shown in panel A1. B. Immunoblot analysis of Anx expression in cortical neurons following the administration of a control or Anx siRNA. C. Bright-field micrographs of cerebral specimens showing the impact of administering a control siRNA, an Anx siRNA, or a combination of AnxA2, AnxA5, AnxA6, and AnxA7 siRNAs on the levels of cerebral CaP deposition (by the Alizarin Red S assay) and infarction (by the TTC assay) at 3 days of ischemic stroke induced at 3 days following siRNA administration. Cerebral specimens were prepared for Alizarin Red S and TTC assays by using the same method described in Fig 9. CTRL: Control. A2, A5, A6, A7: AnxA2, AnxA5, AnxA6, and AnxA7, respectively. The length scale is for all panels. D. Graphic representation of the fractions of the CaP-positive and infarcted cerebrum under various siRNA treatment conditions as in panel C. Red columns: Cerebral CaP deposition. Blue columns: Cerebral infarction. Means and SDs are presented (n = 6). ANOVA p < 0.001 for comparisons in cerebral CaP deposition or infarction across control and various Anx siRNA treatments. Each p value on a column indicates the significant level of difference in cerebral CaP deposition (red in color) or infarction (blue) between treatments with an Anx siRNA and control siRNA. A2-7: Combination of AnxA2, AnxA5, AnxA6, and AnxA7 siRNAs.

Anx/Cap binding capacity to β actin filaments may depend on the relative level of cytosolic monomeric β actin, which can bind to Anxs to competitively block Anx–β actin filament interaction. An increase in the monomeric β actin level prevents Anx/CaP binding to β actin filaments, thereby alleviating Anx/CaP deposition in ischemic neurons (Fig 13).

An increase in the cytosolic $Ca^{2+}$ level is a condition causing Anx/CaP complex formation and binding to β actin filaments. Under physiological conditions, the cytosolic $Ca^{2+}$ level is

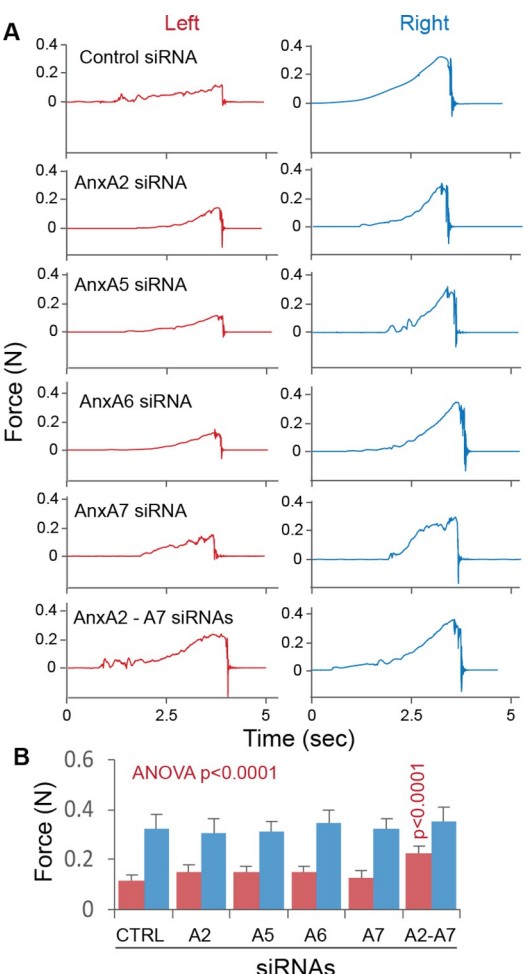

**Fig 12. Mouse forelimb gripping strength in ischemic stroke with administration of control or Anx siRNAs.** A. Experimental records of the left and right mouse forelimb forces at 3 days of ischemic stroke with control or Anx siRNA administration into the cerebrum at 3 days prior to the onset of ischemic stroke. N: Newton. B. Graphic representation of mouse forelimb forces at 3 days of ischemic stroke with the same siRNA treatment conditions as in panel A. The red- and blue-colored columns represent the left and right forelimb forces, respectively. Means and SDs are presented (n = 6). The ANOVA p value is for comparison in the left forelimb force across control (CTRL) and various Anx siRNA treatments. The p value on the A2-A7 red-colored column is for comparison in the left forelimb force between administrations of the control siRNA and the combination of AnxA2, AnxA5, AnxA6, and AnxA7 siRNAs.

about 0.1 μM, which is about 10,000-times lower than that of the extracellular space (~1 mM), establishing a large $Ca^{2+}$ concentration gradient across the cell membrane [35]. Such a cytosolic $Ca^{2+}$ level is insufficient for CaP formation. In ischemic stroke, the rate of $Ca^{2+}$ influx into the neuron increases in response to glutamate exocytosis [22], anaerobic glycolysis-induced lactate acidosis [23], and cell membrane damage. Glutamate exocytosis results in the accumulation of extracellular glutamate, an amino acid activating the ionotropic N-Methyl-D-aspartate (NMDA) and α-amino-3-hydroxy-5-methyl-4-isoxazolepropionic acid (AMPA) receptors that promote $Ca^{2+}$ influx [22]. Lactate acidosis is a consequence of cytosolic $H^+$ accumulation in response to anaerobic glycolytic metabolism, a process causing a chain of cross-membrane ion transport events including $H^+$-$Na^+$ exchange (with $H^+$ efflux and $Na^+$ influx) and $Na^+$-$Ca^{2+}$ exchange (with $Na^+$ efflux and $Ca^{2+}$ influx), resulting in cytosolic $Ca^{2+}$ accumulation [24]. Cell membrane damage causes $Ca^{2+}$ influx along the cross-membrane $Ca^{2+}$

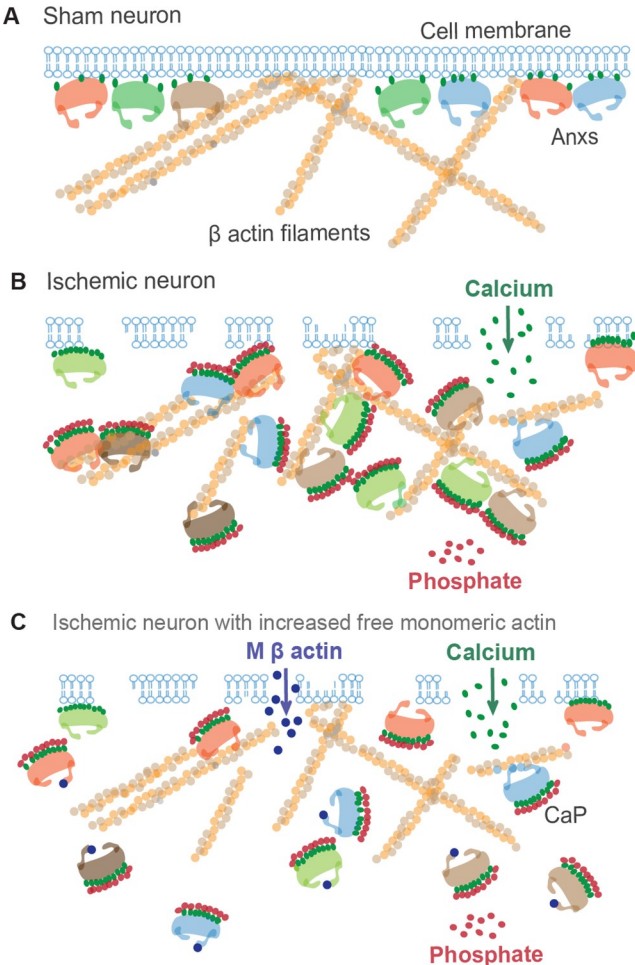

**Fig 13. Mechanisms of Anx-mediated CaP deposition in ischemic neurons.** A. Anx distribution in healthy neurons. B. Anx/CaP formation and deposition to β actin filaments. C. Blockade of Anx binding to β actin filaments by monomeric β actin. PPi: Inorganic pyrophosphate. Pi: Inorganic phosphate. AP: Alkaline phosphatase. M β actin: Monomeric β actin. The green and red dots represents Ca2+ and PO43-, respectively. The double layers of green and red dots represents CaP.

concentration gradient. All these processes result in a rapid cytosolic $Ca^{2+}$ overload. At the same time, alkaline phosphatase is activated in ischemic neurons, as demonstrated in this investigation, to generate excessive $PO_4^{3-}$ from inorganic pyrophosphate [33, 36, 37]. The overload of both $Ca^{2+}$ and $PO_4^{3-}$ ultimately triggers CaP formation. In the presence of Anxs, an increase in $Ca^{2+}$ concentration promotes $Ca^{2+}$ binding to Anxs. The Anx-carried $Ca^{2+}$ can attract $PO_4^{3-}$, resulting in CaP formation on Anx molecules. Anx binding to β actin filaments in turn facilitates CaP deposition and accumulation.

Several lines of evidence support the aforementioned mechanisms: (1) Anxs were able to translocate from the cell membrane to the cytoskeleton (Fig 4), a process associated with CaP deposition in ischemic neurons (Fig 6); (2) Anxs were able to bind to β actin filaments, a process that was blocked with monomeric β actin, suggesting that β actin filaments were an Anx-binding target in the cytoskeleton (Fig 5); (3) Anxs were able to induce the formation of Anx/CaP aggregates in the presence of $Ca^{2+}$ and $PO_4^{3-}$, and the Anx/CaP aggregates were capable of binding to actin filaments on brain specimen sections *in vitro* (Fig 8); (4) Anx administration to the healthy mouse brain caused brain CaP deposition and infarction (Fig 9); (5)

Administration of monomeric β actin was able to mitigate Anx-induced brain CaP deposition and infarction (Fig 9); and (6) Anx gene silencing alleviated brain CaP deposition and infarction in ischemic stroke (Fig 11). It is important to emphasize the following points: Anx administration-induced brain CaP deposition was associated with brain infarction; blockade of Anx-induced Cap deposition by a monomeric β actin treatment mitigated brain infarction; and Anx siRNA-mediated alleviation of neuronal CaP deposition reduced brain infarction. These observations support the concept that Anx-induced neuronal CaP deposition contributes to the development of brain infarction. However, multiple factors can contribute to neuronal death. CaP deposition to β actin filaments may only represent one of these factors.

It should be addressed that neuronal CaP formation has been considered a protective process against neuronal injury in ischemic stroke [38]. A potential mechanism is that CaP formation can deplete free cytosolic $Ca^{2+}$, thereby alleviating $Ca^{2+}$ overload and preventing $Ca^{2+}$-induced activation of intracellular enzymes, including phospholipases, proteases, and nucleases, that can cause cell injury and death [39, 40]. However, a consensus from a larger number of previous investigations is that the presence of intracellular CaP nanoparticles can cause cell injury and death. For instance, the presence of CaP nanoparticles in human vascular smooth muscle cells can cause cell death [41]. Other studies have showed that CaP nanoparticles taken up by cells from substrates can cause cell injury and death [42–48]. Based on these observations, it is possible that the impact of intracellular CaP formation on the cell survivability is dependent on the level of free CaP nanoparticles and the degree of CaP deposition to the intracellular structures. The presence of a low density of free CaP nanoparticles without significant deposition to the intracellular structures may be protective against $Ca^{2+}$ overload and its cytotoxicity, whereas the deposition of a high level of CaP nanoparticles to the intracellular structures, especially those critical to the cell structural integrity and cell functions, may elicit a harmful impact on the cell survivability. β actin filaments are a critical structure supporting neuronal survival as these filaments are involved in many neuronal functions such as signal transduction, maintenance of neuronal integrity, control of synaptic plasticity, and regulation of axonal branching and regeneration [49, 50]. In particular, the dynamic assembly and disassembly processes of β actin is essential to neuronal survival [51]. Exposure to phalloidin, a substance that stabilizes actin filaments and prevents the actin dynamic processes, can cause cell death [52]. CaP deposition to β actin filaments, as observed in this investigation, can interfere with the actin filament functions, thereby contributing to neuronal injury and death.

In this investigation, individual Anxs were introduced to healthy mouse brains to test their effect on brain CaP formation/deposition and infarction. One question is whether the exogenous Anxs can bind to the cell membrane instead of forming Anx/CaP nano-complexes and inducing Anx/CaP deposition to β actin filaments. The relative level of Anx binding to the cell membrane or β actin filaments may depend on the level of exogenous annexins delivered to the cell. At a high level, the Anx binding sites on the inner face of the cell membrane may be fully bound with Anxs, and the excessive exogenous free annexins may form Anx/CaP nano-complexes and bind to β actin filaments. Free Anxs within the cell can attract calcium ions on one side of the disc-like molecule [8, 15, 16], and the Anx-carried calcium ions in turn attract phosphate ions to form Anx/CaP nano-complexes. The other side of an annexin molecule can bind to β actin filaments [8, 17], resulting in Anx/CaP nano-complex deposition to the β actin filaments.

There is a technical point to address. In this investigation, an in situ approach was used to demonstrate the interaction between Anxs and β actin filaments (Fig 8). Freshly prepared healthy mouse brain specimens, instead of ischemic brain specimens, were used for this test. The reason is that annexin–β actin filament interaction occurs in ischemia, making it difficult to distinguish between the newly induced annexin–β actin filament interaction and the

ischemia-induced annexin–β actin filament interaction. In this test, individual annexins were applied to healthy brain sections in the presence of $Ca^{2+}$ and $PO_4^{3-}$, in the presence or absence of monomeric β actin (capable of binding to annexins to block annexin–β actin filament interaction), and in the presence or absence of phalloidin. Outcomes from this test can be used to evaluate whether Anxs bind to β actin filaments to cause CaP deposition to β actin filaments as well as whether the phalloidin-induced stabilization of β actin filaments interferes with Anx binding to the β actin filaments and CaP deposition.

## Supporting information

**S1 Data. Raw data files.**
(ZIP)

**S1 File. Uncropped gel images (for all immunoblot-based figures).**
(DOCX)

## Author Contributions

**Conceptualization:** Shu Q. Liu.

**Data curation:** Shu Q. Liu.

**Formal analysis:** Shu Q. Liu.

**Funding acquisition:** Shu Q. Liu.

**Investigation:** Shu Q. Liu.

**Methodology:** Shu Q. Liu.

**Project administration:** Shu Q. Liu.

**Resources:** Shu Q. Liu.

**Supervision:** Shu Q. Liu.

**Validation:** Shu Q. Liu.

**Visualization:** Shu Q. Liu.

**Writing – original draft:** Shu Q. Liu.

**Writing – review & editing:** Shu Q. Liu, John B. Troy, Jeremy Goldman, Roger J. Guillory.

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
