## [Decision Letter · Decision Letter 0]

29 Apr 2024

PONE-D-24-11802Calcium phosphate formation and deposition in ischemic neuronsPLOS ONE

Dear Dr. Liu,

Thank you for submitting your manuscript to PLOS ONE. After careful consideration, we feel that it has merit but does not fully meet PLOS ONE’s publication criteria as it currently stands. Therefore, we invite you to submit a revised version of the manuscript that addresses the points raised during the review process. Additionally, please provide the uncropped images of all Western blots in supplemental figures. Please also show the molecular weights of all products on all Western blots.

We look forward to receiving your revised manuscript.

Kind regards,

Alexander G Obukhov, Ph.D.

Academic Editor

PLOS ONE

Journal Requirements:

"This work was supported by the National Science Foundation."

"Funded by NSF. "

6. We note that your Data Availability Statement is currently as follows: 

For example, authors should submit the following data: All relevant data are within the manuscript and its Supporting Information files.

7. PLOS ONE now requires that authors provide the original uncropped and unadjusted images underlying all blot or gel results reported in a submission’s figures or Supporting Information files. This policy and the journal’s other requirements for blot/gel reporting and figure preparation are described in detail at https://journals.plos.org/plosone/s/figures#loc-blot-and-gel-reporting-requirements and https://journals.plos.org/plosone/s/figures#loc-preparing-figures-from-image-files. When you submit your revised manuscript, please ensure that your figures adhere fully to these guidelines and provide the original underlying images for all blot or gel data reported in your submission. See the following link for instructions on providing the original image data: https://journals.plos.org/plosone/s/figures#loc-original-images-for-blots-and-gels. 

8. PLOS requires an ORCID iD for the corresponding author in Editorial Manager on papers submitted after December 6th, 2016. Please ensure that you have an ORCID iD and that it is validated in Editorial Manager. To do this, go to ‘Update my Information’ (in the upper left-hand corner of the main menu), and click on the Fetch/Validate link next to the ORCID field. This will take you to the ORCID site and allow you to create a new iD or authenticate a pre-existing iD in Editorial Manager. Please see the following video for instructions on linking an ORCID iD to your Editorial Manager account: https://www.youtube.com/watch?v=_xcclfuvtxQ

9. Please include your full ethics statement in the ‘Methods’ section of your manuscript file. In your statement, please include the full name of the IRB or ethics committee who approved or waived your study, as well as whether or not you obtained informed written or verbal consent. If consent was waived for your study, please include this information in your statement as well. 

Reviewers' comments:

Reviewer's Responses to Questions

**Comments to the Author**

1. Is the manuscript technically sound, and do the data support the conclusions?

Reviewer #1: Partly

Reviewer #2: Yes

2. Has the statistical analysis been performed appropriately and rigorously? 

Reviewer #1: Yes

Reviewer #2: Yes

3. Have the authors made all data underlying the findings in their manuscript fully available?

Reviewer #1: Yes

Reviewer #2: No

4. Is the manuscript presented in an intelligible fashion and written in standard English?

Reviewer #1: Yes

Reviewer #2: Yes

5. Review Comments to the Author

Reviewer #1: Review of Manuscript Number: PONE-D-24-11802

The manuscript entitled “Calcium phosphate formation and deposition in ischemic neurons”, by Shi Q. Liu, John B Troy, Jeremy B Goldman and Roger J. Guillory, describes a potential mechanism for the formation of calcium phosphate precipitates in cortical neurons post-ischemia. This manuscript proposes a compelling mechanism where several annexins form complexes with CaP, in a cytoskeletal dependent manner. However, the argument the author makes would be strengthened by providing more evidence to support and clarify the logic of their proposed mechanism. Some key assumptions require further justification and /or clarification.

Concerns:

1) In the current literature intraneuronal CaP precipitates are considered a post-ischemic protective step of calcium homeostasis. How do the authors reconcile this view with their conclusion that preventing CaP deposition during ischemia can be protective and a potential treatment strategy, particularly while the plasma membrane integrity is still compromised (see Figure 13 C)?

2) Using a biochemistry approach the authors conclude that annexins translocate from the cell membrane to the cytoskeleton. However, they also conclude that when the translocation occurs the plasma membrane integrity is compromised. A leaky plasma membrane is the first step versus irreversible cell death and disassembly of the cytoskeleton. Indeed, recently a paper has shown that ischemic cortical neurons display a dramatic increase in their F-actin content in the somatodendritic region, and that this response is inversely correlated to plasma membrane leakage. Can the authors support their conclusion by using an intracellular in situ approach to monitor the interaction between Annexins and the cellular cytoskeleton post-ischemia? For example PLA?

3) In the manuscript is not explained why when the annexins are added to the healthy mouse brains, this resulted in cerebral CaP deposition and Anx binding to the cytoskeleton. Why in these healthy brains the exogenously added Anx wouldn’t just remain bound to the plasma membrane? Why this manipulation would even cause infarction?

4) In Figure 6 the green actin signal is expected to have the same intensity and distribution in all images, but it doesn’t. Why?

5) Figure 11 is the most convincing piece of evidence that annexins are connected to calcium precipitates and infarct size. However, these findings are counterintuitive based on the current knowledge that knocking down annexins would result in plasma membrane damage and excessive calcium influx/levels.

6) The authors should make sure that all citated papers are correctly referenced. For example Liu 2019 cited on page 28 does not have any mention of CaP depositions.

Conclusion:

Overall, this manuscript represents an informative new set of experiments that provides insights in the connection between Annexins and CaP precipitates. However, by addressing the limitations outlined above, the authors could strengthen the overall impact of their work.

Reviewer #2: This study aims to establish that CaP deposition occurs early after an ischemic stroke and that it participates in the associated injury. The authors propose a mechanism after which specific annexins are crucial for CaP deposition via binding to ß actin filaments.

First, the study tries to establish the presence of CaP deposits after ischemia in a mouse model sytem. Then the expression and subcelluar distribution of annexins is investigated by Western blot followed by an analysis of binding of annexins to ß actin. The alteration in distribution of actin and annexins was investigated by immunofluorescence after stroke induction. Furthermore, actin and annexin dependent CaP deposition was investigated. As a behavioral measure, grip strength was investigated and compared under the different conditions for the lesioned and intact site. Finally, the specific effect of particular annexins was investigated by blocking annexin expression by siRNAs.

The experiments follow a logical outline and are conclusive. However, the data presented do not convince in all aspects:

-the pictures in Figure 3A which shall illustrate an increase in alkaline phosphatase activity are not convincing as the increased staining is hard to distinguish from the background.

- in Fig 6, the alteration in distribution of actin and annexins is illustrated by single pictures (representative?) but not quantified.

- it remains unclear whether the proposed CaP deposits form in intact cells or only after disruption of the cell membrane, as indicated in Fig 13. If Ca2+ entry and CaP deposition are a consequence of cell membrane damage, prevention of CaP deposition for treatment as proposed in the discussion would be too late.

The individual data points underlying the summary statistics are not available.

6. PLOS authors have the option to publish the peer review history of their article (what does this mean?). If published, this will include your full peer review and any attached files.

Reviewer #1: **Yes: **Barbara Calabrese

Reviewer #2: No

---

## [Author Response · Author response to Decision Letter 0]

19 Sep 2024

Response to the Reviewers

The authors would like to thank the reviewers for providing thoughtful and informative comments to the manuscript. The manuscript has been revised in accordance to the reviewers’ comments. Please note that all revisions are marked in blue in the Word document. The comments are addressed here. 

Response to Reviewer 1

The manuscript entitled “Calcium phosphate formation and deposition in ischemic neurons”, by Shi Q. Liu, John B Troy, Jeremy B Goldman and Roger J. Guillory, describes a potential mechanism for the formation of calcium phosphate precipitates in cortical neurons post-ischemia. This manuscript proposes a compelling mechanism where several annexins form complexes with CaP, in a cytoskeletal dependent manner. However, the argument the author makes would be strengthened by providing more evidence to support and clarify the logic of their proposed mechanism. Some key assumptions require further justification and /or clarification.

Concerns:

1) In the current literature intraneuronal CaP precipitates are considered a post-ischemic protective step of calcium homeostasis. How do the authors reconcile this view with their conclusion that preventing CaP deposition during ischemia can be protective and a potential treatment strategy, particularly while the plasma membrane integrity is still compromised (see Figure 13 C)?

Response: Intracellular CaP precipitation may represent a protective mechanism against Ca2+ overload and its cytotoxicity in ischemic neurons [Ramonet D, Pugliese M, Rodríguez MJ, de Yebra L, Andrade C, Adroer R, Ribalta T, Mascort J, Mahy N. Calcium precipitation in acute and chronic brain diseases. J Physiol Paris. 96(3-4):307-312, 2002.]. However, a consensus from a larger number of previous investigations is that the presence of intracellular CaP nanoparticles can cause cell injury and death. For instance, the presence of CaP nanoparticles in human vascular smooth muscle cells can cause cell death [Ewence AE, Bootman M, Roderick HL, Skepper JN, McCarthy G, Epple M, Neumann M, Shanahan CM, Proudfoot D. Calcium phosphate crystals induce cell death in human vascular smooth muscle cells: a potential mechanism in atherosclerotic plaque destabilization. Circ Res. 103(5):e28-34, 2008]. Other studies support this observation [Epple M. Review of potential health risks associated with nanoscopic calcium phosphate. Acta Biomaterialia 77, 1-14, 2018; Z.L. Shi, X. Huang, Y.R. Cai, R.K. Tang, D.S. Yang. Size effect of hydroxyapatite nanoparticles on proliferation and apoptosis of osteoblast-like cells. Acta Biomater., 5 (2009), pp. 338-345; Z. Xu, C. Liu, J. Wei, J. Sun. Effects of four types of hydroxyapatite nanoparticles with different nanocrystal morphologies and sizes on apoptosis in rat osteoblasts. J. Appl. Toxicol., 32 (2012), pp. 429-435; H. Turkez, M.I. Yousef, E. Sonmez, B. Togar, F. Bakan, P. Sozio, A.D. Stefano. Evaluation of cytotoxic, oxidative stress and genotoxic responses of hydroxyapatite nanoparticles on human blood cells. J. Appl. Toxicol., 34 (2014), 373-379; E. Sonmez, I. Cacciatore, F. Bakan, H. Turkez, Y.I. Mohtar, B. Togar, A.D. Stefano. Toxicity assessment of hydroxyapatite nanoparticles in rat liver cell model in vitro. Hum. Exp. Toxicol., 35 (2016), 1073-1083; Y. Jin, X.L. Liu, H.F. Liu, S.Z. Chen, C.Y. Gao, K. Ge, C.M. Zhang, J.C. Zhang. Oxidative stress-induced apoptosis of osteoblastic MC3T3-E1 cells by hydroxyapatite nanoparticles through lysosomal and mitochondrial pathways. RSC Adv., 7 (2017), 13010-13018; Y. Dautova, D. Kozlova, J.N. Skepper, M. Epple, M.D. Bootman, D. Proudfoot. Fetuin-A and albumin alter cytotoxic effects of calcium phosphate nanoparticles on human vascular smooth muscle cells. PLoS One, 9 (2014), e97565]. Based on these observations, it is possible that the impact of CaP formation on cell survivability may depend on the level of free CaP nanoparticles and the degree of CaP deposition to intracellular structures. The presence of a low density of free CaP nanoparticles without significant deposition to intracellular structures may be protective against Ca2+ overload and cytotoxicity; whereas the deposition of a high level of CaP nanoparticles to intracellular structures, especially those critical to cell signaling and integrity, may elicit a harmful impact on cell survivability. 

β actin filaments support many neuronal functions such as signal transduction, maintenance of cell integrity, regulation of synaptic plasticity, and control of axonal branching and regeneration. In particular, the dynamic assembly and disassembly processes of β actin is essential to neuronal survival [Patrinostro X, Roy P, Lindsay A, Perrin BJ. Essential nucleotide- and protein-dependent functions of Actb/β-actin. PNAS 115 (31) 7973-7978, 2018]. Exposure to phalloidin, a substance that stabilizes actin filaments and prevents the dynamic actin processes, can cause cell death [Cooper, J. A. Effects of cytochalasin and phalloidin on actin. J. Cell. Biol. 105, 1473–1478, 1987]. CaP deposition on β actin filaments, as observed in this investigation, can interfere with these dynamic actin processes, thereby contributing to neuronal injury and death. The points above are addressed in the revised manuscript (page 19 – 20 of the Word document). 

2) Using a biochemistry approach the authors conclude that annexins translocate from the cell membrane to the cytoskeleton. However, they also conclude that when the translocation occurs the plasma membrane integrity is compromised. A leaky plasma membrane is the first step versus irreversible cell death and disassembly of the cytoskeleton. Indeed, recently a paper has shown that ischemic cortical neurons display a dramatic increase in their F-actin content in the somatodendritic region, and that this response is inversely correlated to plasma membrane leakage. Can the authors support their conclusion by using an intracellular in situ approach to monitor the interaction between Annexins and the cellular cytoskeleton post-ischemia? For example PLA?

Response: In this investigation, an in situ approach was used to demonstrate the interaction between annexins and β actin filaments (Fig. 8 in the manuscript). Freshly prepared healthy mouse brain specimens, instead of ischemic brain specimens, were used for this test. The reason is that annexin – β actin filament interaction occurs in ischemia, making it difficult to distinguish between the newly induced annexin – β actin filament interaction and the ischemia-induced annexin – β actin filament interaction. 

In this test, individual annexins were added to brain sections in the presence of Ca2+ and PO43-, in the presence or absence of an annexin, in the presence or absence of monomeric β actin (capable of binding to annexins to block annexin – β actin filament interaction), and in the presence or absence of phalloidin. The outcomes can be used to evaluate whether annexins bind to β actin filaments to cause CaP deposition to β actin filaments as well as whether the phalloidin-induced stabilization of β actin filaments interferes with annexin binding to the β actin filaments and CaP deposition. 

Please note that the total level of β actin was not changed from sham control to ischemic stroke as evaluated by immunoblot analysis in this investigation (Figure 4 in the manuscript). 

3) In the manuscript is not explained why when the annexins are added to the healthy mouse brains, this resulted in cerebral CaP deposition and Anx binding to the cytoskeleton. Why in these healthy brains the exogenously added Anx wouldn’t just remain bound to the plasma membrane? Why this manipulation would even cause infarction?

Response: The relative level of annexin binding to the cell membrane or β actin filaments may depend on the quantity of exogenous annexins delivered to cells. At a large quantity, the annexin binding sites on the inner face of the cell membrane may all be bound with annexins, and the excessive exogenous free annexins may bind to β actin filaments. Free annexins within the cell can attract calcium ions, which in turn attract phosphate ions to form CaP nanostructures on one side of a disk-like annexin molecule. The other side of the annexin molecule can bind to β actin filaments, resulting in annexin/CaP complex deposition onto the β actin filaments or β actin filament calcification. This process contributes to neuronal death and brain infarction. These points are addressed in the revised manuscript (page 20 of the Word document).

4) In Figure 6 the green actin signal is expected to have the same intensity and distribution in all images, but it doesn’t. Why?

Response: This could be a result of variations in experimental preparations and locations for specimen collection among different mice. 

5) Figure 11 is the most convincing piece of evidence that annexins are connected to calcium precipitates and infarct size. However, these findings are counterintuitive based on the current knowledge that knocking down annexins would result in plasma membrane damage and excessive calcium influx/levels.

Response: It is possible that knocking down annexins could result in plasma membrane damage and excessive calcium influx. Even though additional injury may be induced by annexin siRNA delivery, silencing of the annexin genes resulted in a significant alleviation of CaP deposition and brain infarction, supporting the role of annexins in ischemic neuron calcification and infarction. 

6) The authors should make sure that all citated papers are correctly referenced. For example Liu 2019 cited on page 28 does not have any mention of CaP depositions.

Response: This error has been corrected.

Conclusion: Overall, this manuscript represents an informative new set of experiments that provides insights in the connection between Annexins and CaP precipitates. However, by addressing the limitations outlined above, the authors could strengthen the overall impact of their work.

Response: Thanks. The comments have been addressed to our best knowledge. 

Response to Reviewer 2

This study aims to establish that CaP deposition occurs early after an ischemic stroke and that it participates in the associated injury. The authors propose a mechanism after which specific annexins are crucial for CaP deposition via binding to ß actin filaments.

First, the study tries to establish the presence of CaP deposits after ischemia in a mouse model system. Then the expression and subcelluar distribution of annexins is investigated by Western blot followed by an analysis of binding of annexins to ß actin. The alteration in distribution of actin and annexins was investigated by immunofluorescence after stroke induction. Furthermore, actin and annexin dependent CaP deposition was investigated. As a behavioral measure, grip strength was investigated and compared under the different conditions for the lesioned and intact site. Finally, the specific effect of particular annexins was investigated by blocking annexin expression by siRNAs.

The experiments follow a logical outline and are conclusive. However, the data presented do not convince in all aspects:

-the pictures in Figure 3A which shall illustrate an increase in alkaline phosphatase activity are not convincing as the increased staining is hard to distinguish from the background.

Response: Although changes in alkaline phosphatase activity are not apparent enough to be recognized by naked eyes, optical measurements have shown significant changes from the images shown in the manuscript. 

- in Fig 6, the alteration in distribution of actin and annexins is illustrated by single pictures (representative?) but not quantified.

Response: Yes, the images are representative, but all samples for each group showed similar distributions in annexins. The outcomes were not quantified because it is difficult to quantify the distributions of annexins and β actin filaments in neurons by an imaging approach.

- it remains unclear whether the proposed CaP deposits form in intact cells or only after disruption of the cell membrane, as indicated in Fig 13. If Ca2+ entry and CaP deposition are a consequence of cell membrane damage, prevention of CaP deposition for treatment as proposed in the discussion would be too late.

Response: Multiple factors contribute to cell death. Cell membrane damage is one of these factors, and β actin filament calcification may represent another cell death-inducing factor. Preventing annexin-induced β actin filament calcification may partially alleviate neuronal death. This point is addressed in the revised manuscript (page 19 of the revised manuscript).

The individual data points underlying the summary statistics are not available.

Response: All raw data are now available in the Supplemental Information section.

---

## [Decision Letter · Decision Letter 1]

10 Oct 2024

PONE-D-24-11802R1Calcium phosphate formation and deposition in ischemic neuronsPLOS ONE

Dear Dr. Liu,

Thank you for submitting your manuscript to PLOS ONE. After careful consideration, we feel that your manuscript has merit, but it still requires additional revisions to meet PLOS ONE’s publication criteria. Specifically, Reviewer 2 raised several serious concerns. Therefore, we invite you to submit a revised version of the manuscript that addresses the concerns of Reviewer 2.

We look forward to receiving your revised manuscript.

Kind regards,

Alexander G Obukhov, Ph.D.

Academic Editor

PLOS ONE

Reviewers' comments:

Reviewer's Responses to Questions

**Comments to the Author**

1. If the authors have adequately addressed your comments raised in a previous round of review and you feel that this manuscript is now acceptable for publication, you may indicate that here to bypass the “Comments to the Author” section, enter your conflict of interest statement in the “Confidential to Editor” section, and submit your "Accept" recommendation.

Reviewer #1: All comments have been addressed

Reviewer #2: (No Response)

2. Is the manuscript technically sound, and do the data support the conclusions?

Reviewer #1: (No Response)

Reviewer #2: Partly

3. Has the statistical analysis been performed appropriately and rigorously? 

Reviewer #1: (No Response)

Reviewer #2: Yes

4. Have the authors made all data underlying the findings in their manuscript fully available?

Reviewer #1: (No Response)

Reviewer #2: Yes

5. Is the manuscript presented in an intelligible fashion and written in standard English?

Reviewer #1: (No Response)

Reviewer #2: Yes

6. Review Comments to the Author

Reviewer #1: (No Response)

Reviewer #2: My concerns were not sufficiently addressed by the response of the authors.

- Selecting areas for quantification in Fig 3 appears very arbitrary given the high background.

- I do not see, why quantification for Fig 6 distribution of actin and annexins poses a major problem.

- Prevention of CaP deposition for treatment as proposed in the discussion would be too late as the cell membranes are already damaged. The response to this criticism does not settle the point.

- The primary data were now supplied as supplement.

7. PLOS authors have the option to publish the peer review history of their article (what does this mean?). If published, this will include your full peer review and any attached files.

Reviewer #1: No

Reviewer #2: No

---

## [Author Response · Author response to Decision Letter 1]

25 Nov 2024

Response to Reviewer 2

The authors would like to thank the reviewer for providing insightful comments. The manuscript has been revised based on the reviewer’s concerns. All changes are marked in blue in the revised manuscript. The reviewer’s concerns are addressed here.

Reviewer’s concern: My concerns were not sufficiently addressed by the response of the authors.

Response: Sorry that the authors did not address fully the reviewer’s concerns. In this revision, these concerns have been carefully addressed and substantial changes have been made accordingly. 

Reviewer’s concern: - Selecting areas for quantification in Fig 3 appears very arbitrary given the high background.

Response: Figure 3A, B and associated data have been removed from the revised manuscript. Results from the remaining p-nitrophenyl phosphate (pNPP) test would be sufficient to support the observation that the activity of alkaline phosphatase was increased in the cortex in response to ischemic stroke.

Reviewer’s concern: - I do not see, why quantification for Fig 6 distribution of actin and annexins poses a major problem.

Response: The distribution of annexins in reference to that of the β actin filaments has been measured and presented in Fig. 6B. 

Reviewer’s concern: - Prevention of CaP deposition for treatment as proposed in the discussion would be too late as the cell membranes are already damaged. The response to this criticism does not settle the point.

Response: The authors are in agreement with the reviewer. The statement on “prevention of CaP deposition for the treatment of myocardial infarction” has been removed from the Abstract and Discussion sections. The revised manuscript focusses primarily on the molecular mechanisms of calcium phosphate formation and deposition in ischemic neurons. 

Reviewer’s concern: - The primary data were now supplied as supplement.

Response: Thanks.

---

## [Decision Letter · Decision Letter 2]

20 Dec 2024

Calcium phosphate formation and deposition in ischemic neurons

PONE-D-24-11802R2

Dear Dr. Liu,

We’re pleased to inform you that your manuscript has been judged scientifically suitable for publication and will be formally accepted for publication once it meets all outstanding technical requirements.

Kind regards,

Alexander G Obukhov, Ph.D.

Academic Editor

PLOS ONE

Reviewers' comments:

Reviewer's Responses to Questions

**Comments to the Author**

1. If the authors have adequately addressed your comments raised in a previous round of review and you feel that this manuscript is now acceptable for publication, you may indicate that here to bypass the “Comments to the Author” section, enter your conflict of interest statement in the “Confidential to Editor” section, and submit your "Accept" recommendation.

Reviewer #2: All comments have been addressed

2. Is the manuscript technically sound, and do the data support the conclusions?

Reviewer #2: (No Response)

3. Has the statistical analysis been performed appropriately and rigorously? 

Reviewer #2: (No Response)

4. Have the authors made all data underlying the findings in their manuscript fully available?

Reviewer #2: (No Response)

5. Is the manuscript presented in an intelligible fashion and written in standard English?

Reviewer #2: (No Response)

6. Review Comments to the Author

Reviewer #2: (No Response)

7. PLOS authors have the option to publish the peer review history of their article (what does this mean?). If published, this will include your full peer review and any attached files.

Reviewer #2: No

---

## [Editor Report · Acceptance letter]

7 Jan 2025

PONE-D-24-11802R2 

PLOS ONE

Dear Dr. Liu, 

I'm pleased to inform you that your manuscript has been deemed suitable for publication in PLOS ONE. Congratulations! Your manuscript is now being handed over to our production team.

Kind regards, 

on behalf of

Dr. Alexander G Obukhov 

Academic Editor

PLOS ONE